# The Restored Premolars Biomechanical Behavior: FEM and Experimental Moiré Analyses

**Jose Luis Valin Rivera** [1,*] **, Edison Gonçalves** [2] **, Paulo Vinicius Soares** [3] **, Giovana Milito** [3] **, Jorge Octavio Ricardo Perez** [4] **, Guillermo Francisco Palacios Roque** [4] **, Meylí Valin Fernández** [5] **, Henry Figueredo Losada** [6] **, Fabrícia Araújo Pereira** [7] **, Gilberto Garcia del Pino** [8] **and Alexander Rodriguez Soto** [1]

1 Escuela de Ingeniería Mecánica, Pontificia Universidad Católica de Valparaíso, Quilpué 2430000, Chile; angel.rodriguez@pucv.cl
2 Polytechnic School, University of Sao Paulo, São Paulo 05508-010, Brazil; edison@usp.br
3 Faculty of Dentistry, University of Uberlandia, Uberlândia 38405-266, Brazil; paulovsoares@foufu.ufu.br (P.V.S.); giovanamilito@gmail.com (G.M.)
4 Faculty of Natural Science, University of Oriente, Santiago de Cuba 90100, Cuba; jrperez@cnt.uo.edu.cu (J.O.R.P.); palaciosg226@gmail.com (G.F.P.R.)
5 Department of Mechanical Engineering (DIM), Faculty of Engineering (FI), University of Concepción, Concepción 4030000, Chile; mvalin@udec.cl
6 Faculty of Engineering, University of the Republic, Montevideo 11200, Uruguay; henryf@fing.edu.uy
7 Department of Dentistry, University of Brasilia, Brasília 70910-900, Brazil; fabricia_pereira@hotmail.com
8 College of Technology, State University of Amazonas, Manaus 69850-000, Brazil; gpino@uea.edu.br
* Correspondence: jose.valin@pucv.cl

**Abstract:** This study applied the finite element method (FEM) and the moiré strip projection method to evaluate the biomechanical behavior of healthy and endodontic-treated premolar teeth. The finite element method and the moiré strip projection method were applied to evaluate the influence of restored materials in association with cervical lesions and were considered as strain estimates for a tooth sample with 21 units, under loads of 25, 50, 75, and 100 N, frontal and oblique applied. The focused cases were: tooth H healthy; tooth A-MOD amalgam; tooth AL-MOD amalgam + lesion; tooth ALR-MOD amalgam + injury restored; tooth R-MOD resin; tooth RL-MOD resin + lesion; tooth RLR-MOD resin + injury restored. The results obtained by FEM simulation can be considered perfectly validated by the results presented by the experimental moiré projection analysis, demonstrating that the FEM numerical analysis can be used to evaluate the biomechanical behavior of healthy and endodontically treated teeth. Developing an alternative protocol to generate FEM three-dimensional models will lead to a ready and inexpensive tool since there is no need for costly equipment for tooth extraction prognosis.

**Keywords:** fringe projection; premolars teeth; interferometry; FEM

## 1. Introduction

The loss of dental structure due to caries, trauma, cavity preparation, and endodontic treatment has a negative impact on tooth fracture resistance [1–3]. The commonly proposed functional reconstructions for premolars with extensive structure reduction have been the direct adhesive techniques, using composite resins due to the high adhesion capacity to the dental structures [4] and to seek minimal dental damage during the repair preparations [5]. Other types of treatments, generally less conservative, are available to treat posterior teeth using indirect treatments, nowadays generally performed with CAD/CAM technology [6,7]. Endodontic treatments are not considered complete and successful until definitive coronary restoration is performed [8–10].

Endodontic repair that restores the lost tooth structure's biomechanical behavior would allow for a good lifetime tooth prognosis under masticatory loads. Some authors

recommend the use of amalgam fillings and other resinous materials fillings [11,12]. Dental amalgams, due to their favorable mechanical properties, are widely used as restorative materials, since they use a simple restorative technique and have a good predictable clinical performance. However, they present some disadvantages, such as low dental structure adhesion and a reduced strain resistance of the remaining dental structure. Amalgam-restored teeth tend to fracture due to the multiplication of micro cracks under alternated fatigue loads. The composites, as restoring materials, have mechanical properties, such as elastic modulus similar to the tooth structure, that allow better distributions of stresses and strains throughout the material and the remaining tooth structure. Even so, they have not been successful as an alternative to amalgams to restore non-vital teeth [13]. The loss of tooth structure in the cervical region is very common [14,15]. Cervical tooth structure wear occurs near the cement enamel junction and affects mainly premolars, mainly as a result of the stress concentration in the cusps of the occlusal region.

This study evaluates the influence of different types of restorative material in premolar cavities, associated with the presence of cervical lesions under the action of axial and oblique loads. The biomechanical behaviors of endodontic-treated premolars are analyzed by using numerical, computational, and experimental non-destructive techniques. In order to fulfill this purpose, frontal and oblique loads were applied in the values of 25, 50, 75, and 100 N, in both experimental and numerical-computational biomechanical behavior evaluations. The aforementioned loads were selected in order to compare the results obtained with previous work developed by the authors of this work, in which the extensometry technique was applied.

## 2. Materials and Methods

### 2.1. Selection and Preparation of Tooth Samples

From the Faculty of Dentistry Bank of Teeth, University of Sao Paulo, twenty-one (21) healthy human pre-upper molars (1st and 2nd premolars), with an indication for extraction, were selected. This selection was approved by the Research Ethics Committee: Final Analysis no. 520/11 of the Research Ethics Committee for the Protocol Registration CEP/UFU No. 171/11 and Final Analysis 372/11 of the Research Ethics Committee for the Protocol Registration CEP/UFU 065/11 the Research Ethics Committee of the Federal University of Uberlandia.

The teeth, after the coronary surfaces were measured, were divided into seven groups with three teeth each, which received the following treatments: (1) endodontic treatment; (2) mesial-distal-occlusal cavity (MOD) preparation; (3) mod resin composite/amalgam restoration to first amalgam group and restoration in the second group; (4) cervical ripening; (5) cervical resin composite/amalgam restoration. Thus, the teeth were classified according to the following nomenclature, which will be used in development work: tooth H healthy; tooth A-MOD amalgam; tooth AL-MOD amalgam + lesion; tooth ALR-MOD amalgam + injury restored; tooth R-MOD resin; tooth RL-MOD resin + lesion; tooth RLR-MOD resin + injury restored [16,17].

The teeth were prepared with endodontic access and different restorative materials. The cavities made in the teeth were the mesial-distal occlusion-type (MOD) and cervical preparation (Figure 1). The cavities were standardized following the standard thickness of the diamond bur 1151 (KG Sorensen, Barueri, SP, Brazil). The wear amount considered was occlusal box with an opening of 3.65 mm and the proximal box with a depth of 2.0 mm. The inclination of the occlusal and proximal boxes was also defined by the diamond stylus configuration and therefore parallel (Figure 2). Figure 3 shows the prepared proximal boxes, with special attention to the detail of the rounding of angles. Figure 4 shows restoration MOD models with cervical cavity and restoration MOD and cervical restoration, both trying to imitate the shape of the coronary healthy tooth.

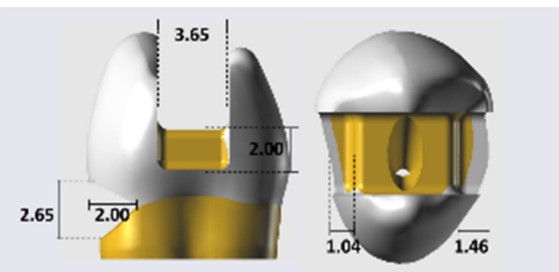

**Figure 1.** Cavities made in the models of the mesial-distal occlusion-type (MOD).

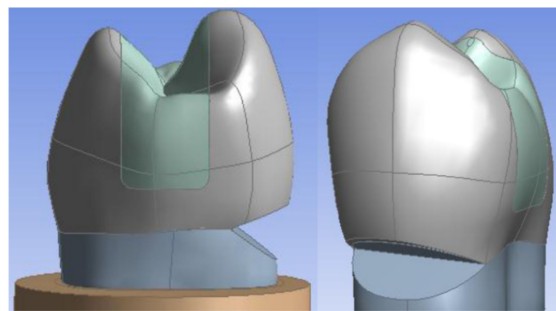

**Figure 2.** Cavities made in models of type cervical preparation.

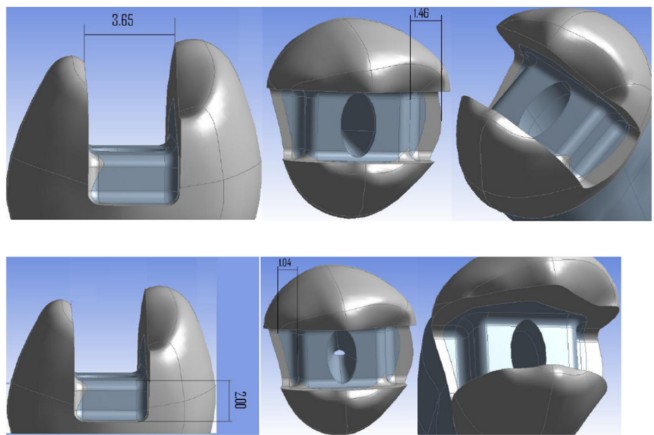

**Figure 3.** Making the proximal boxes, observe the detail of the rounding of angles. Amount of wear: occlusal box with a 3.65 mm box opening, a proximal box with a depth of 2.0 mm.

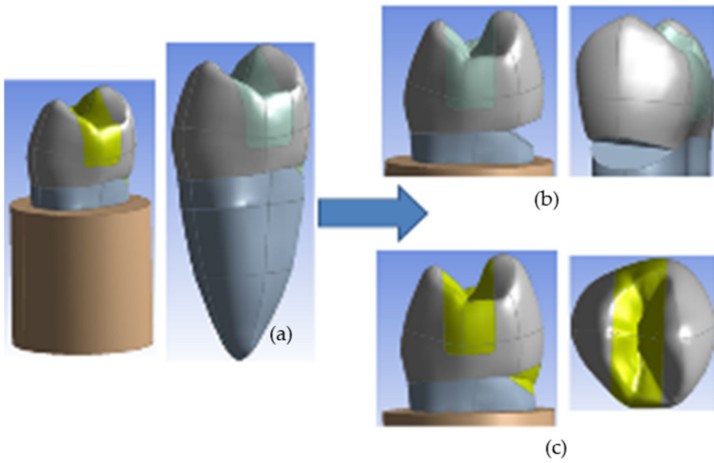

**Figure 4.** Restoration: (**a**) MOD, (**b**) models with a cervical cavity, and (**c**) restoration MOD and cervical restoration.

### 2.2. Finite Element Analysis

Finite element analysis has been used in other studies to determine stress and strain distribution. In [18] the hypothesis that restoration of class II mesio-occlusal-distal (MOD) cavities can be strengthened through judicious choice of restoration geometry and material properties was proven. Research by Shah et al. 2021 [19] used computational analysis of prepared crowned teeth to differentiate the possibility of using porcelain material for typical clinical conditions.

In the present work, the FEM analysis was developed using the computational code ANSYS Academic. The establishment of the physical model was performed using the mean dimensions of the premolars selected set and favorable coronary anatomy as a standard for the healthy and the repaired cases. Thereafter, with the help of a scanner, the outer tooth surface was generated, using a 0.2 mm contact tip trace calibration. The external geometry data was stored in the type "*.STL" computer files (Stereolithographic), generated by the scanner system. These files were exported to Bio-CAD modeling software (Computer Assisted Design; Rhino3D 4.0, rhinoceros USA) to produce the standard three-dimensional model of the healthy tooth. The repaired main dental regions were generated by using the software called NURBS (Non-Uniform Rational Basis Spline) suitable for modeling complex geometries. Figure 5 shows: (a) Model STL computer connected to the scanner system; (b) Software Bio-modeling CAD (Computer Assisted Design; Rhino3D 4.0, Rhinoceros, New York, NY, USA) NURBS surfaces (Non-Uniform Rational Basis Spline); (c) three-dimensional model generation that serves as a standard healthy tooth. Figure 6 shows the model generation of healthy teeth based on the main anatomical landmarks and with the help of a specific tool program (NURBS surfaces). Finally, Figure 7 shows the volumes of the internal and external structures of the reference teeth.

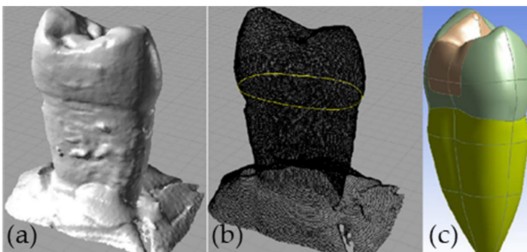

**Figure 5.** (**a**) Model STL, (**b**) Bio-CAD modeling, and (**c**) three-dimensional model generation.

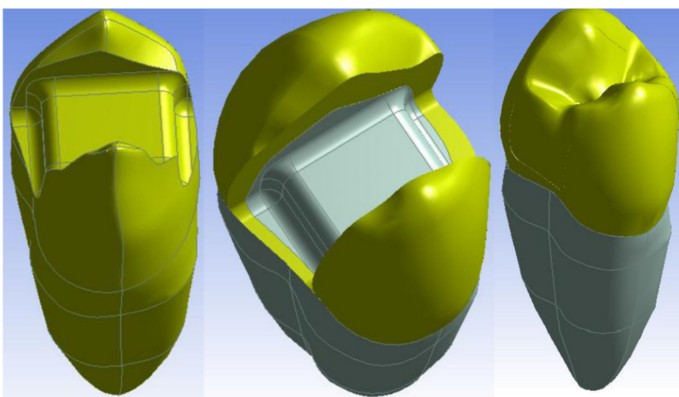

**Figure 6.** Model generation is healthy, based on the main anatomical landmarks and with the help of a specific tool program (NURBS surfaces).

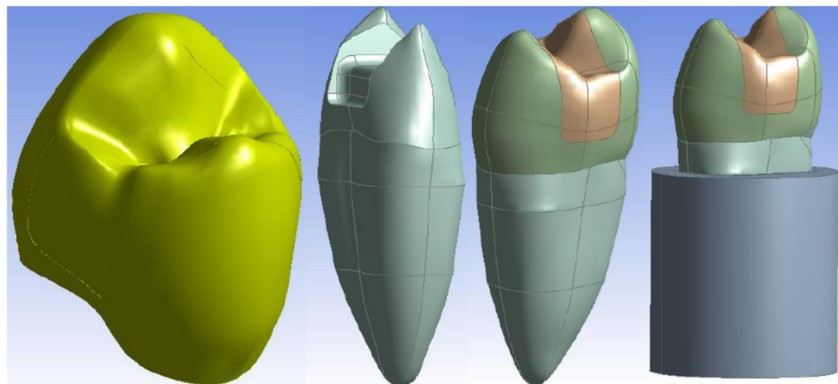

**Figure 7.** Generation of volumes of internal and external structures of the reference tooth.

The geometric models were exported to ANSYS Academic preprocessing software and, for each model, meshes were generated using tetrahedral elements of the linear type, with a total of 68,860 nodes and 40,358 elements. The generated meshes on the models with MOD cavity [20], and endodontic access plus restoration on models with cervical [21], and corresponding restoration cavity are shown, respectively, in Figures 8 and 9. As shown in Figure 10, the boundary conditions were defined for perfectly simulating contacts between the fixing structure (gray cylinder) and models with MOD cavities restored with composite resin.

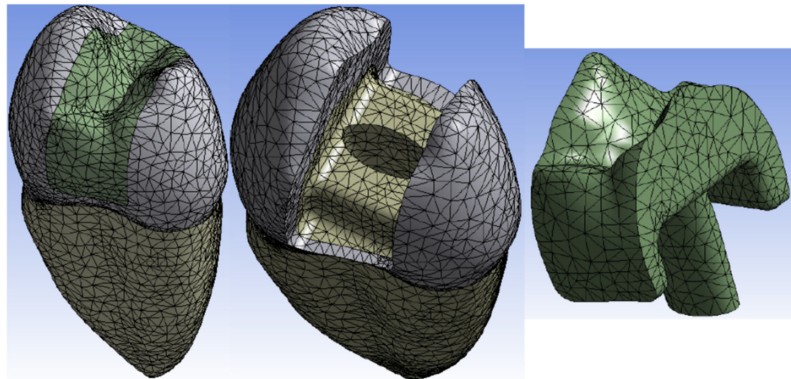

**Figure 8.** Mesh generated on models with MOD cavity and endodontic access with the corresponding restoration.

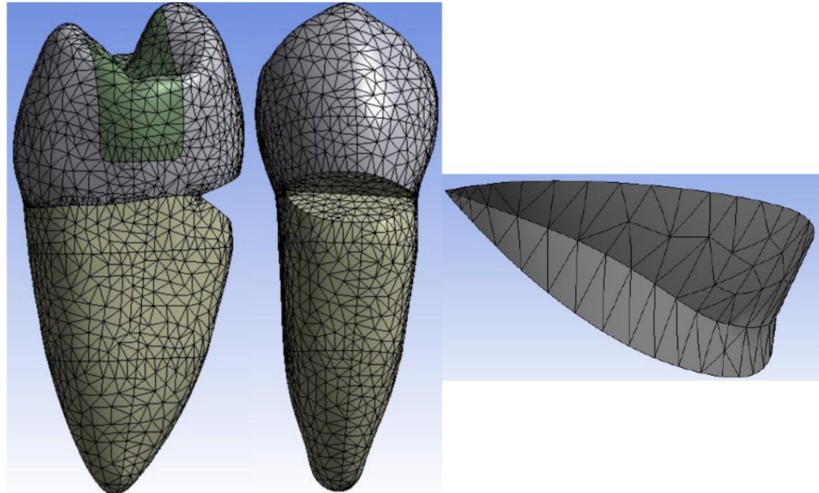

**Figure 9.** Mesh generated on models with cervical and corresponding restoration cavity.

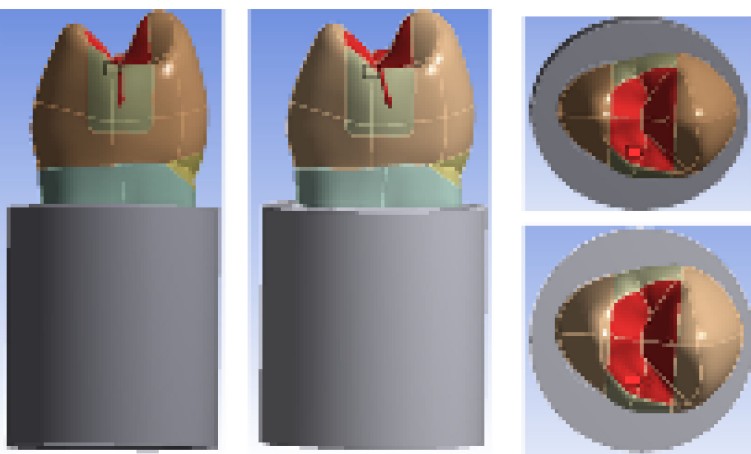

**Figure 10.** Contact area simulation of applied loads.

The tetrahedral model consists of 68,860 nodes and 40,358 elements. The analysis employed was a structural, linear, and elastic type for all such structures were isotropic, linear, and homogeneous. For this, it was necessary to obtain the elastic moduli and Poisson's ratios of the structures which characterize each model, which are described below in Table 1.

**Table 1.** Mechanical properties applied models.

| Material | Modulus of Elasticity E (Mpa) | Poisson's Ratio ($\upsilon$) |
|---|---|---|
| Enamel | $46.8 \times 10^3$ | 0.30 |
| Dentine | $18.0 \times 10^3$ | 0.31 |
| Polystyrene resin | $13.5 \times 10^3$ | 0.30 |
| Composite resin | $16.6 \times 10^3$ | 0.24 |
| Amalgam | $15.0 \times 10^3$ | 0.30 |

For the teeth modeled with FEM, the periodontal tissue PDL was not considered to be maintained under the same conditions as in the moiré experiments.

Axial and oblique loads of 25 N, 50 N, 75 N, and 100 N were applied in the red area shown in Figure 10. The models were exported to the software processing core (Ansys Academic). For analysis of the results, criteria for normal direction deformation were used. The materials were considered homogeneous, isotropic, and with linear-elastic behavior.

The FEM numerical simulation estimated the normal elastic strain (for *X*-, *Y*-, and *Z*-axis) for all the specified cases mentioned before. Figure 11 shows, as an example, only for loading 100 N, a typical 3D color strain representation on the *Z*-axis direction.

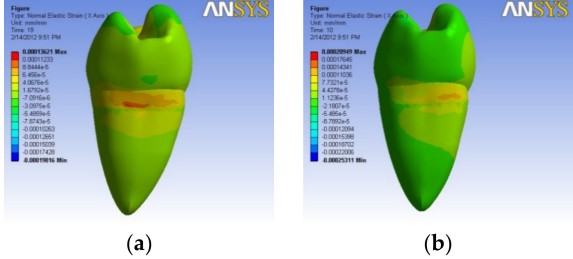

(**a**)                    (**b**)

**Figure 11.** *Cont*.

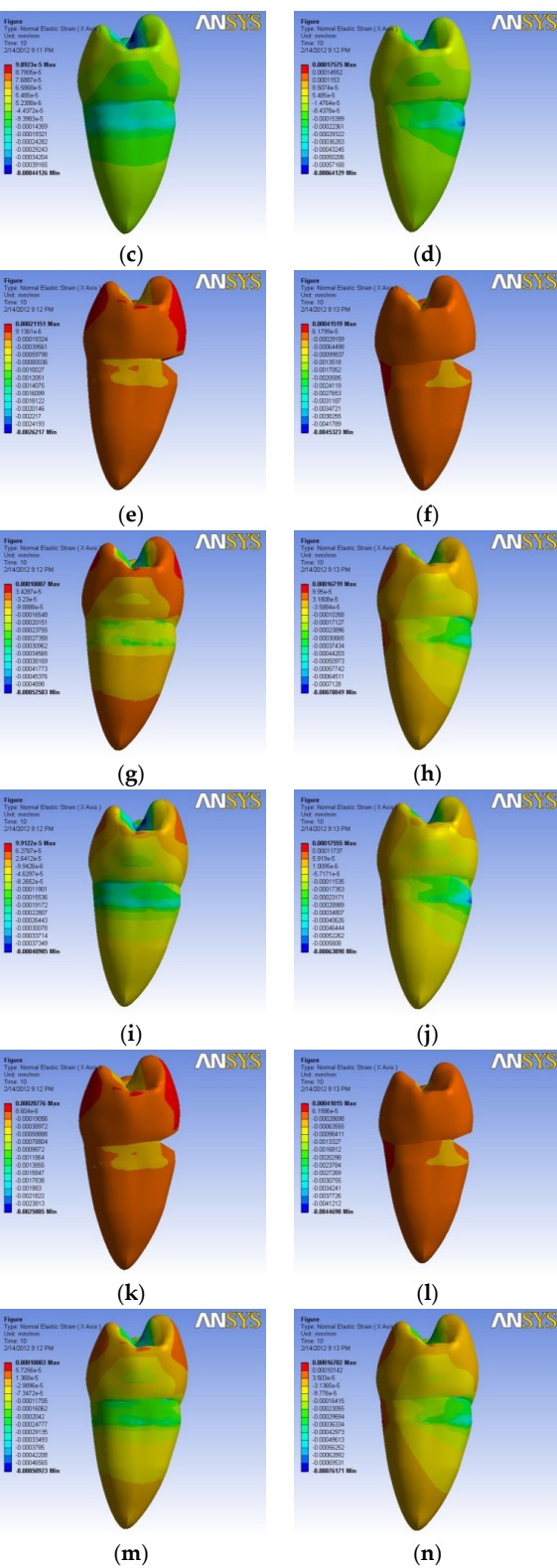

**Figure 11.** Typical 3D color strain representation on the Z-axis direction for 100 N loading: (**a**) tooth H healthy; (**b**) tooth H healthy load oblique 15°; (**c**) tooth A-MOD amalgam; (**d**) tooth A-MOD amalgam load oblique 15°; (**e**) tooth AL-MOD amalgam + lesion; (**f**) tooth AL-MOD amalgam + lesion oblique load 15°; (**g**) tooth ALR-MOD amalgam + restore lesion; (**h**) tooth ALR-MOD amalgam + restore lesion load oblique 15°; (**i**) tooth R-MOD resin; (**j**) tooth RMOD resin load oblique 15°; (**k**) tooth RL-MOD resin + lesion; (**l**) tooth RL-MOD resin + lesion oblique load 15°; (**m**) tooth RLR-MOD resin + restore lesion; (**n**) tooth RLR-MOD resin + lesion oblique load 15°.

The FEM technique strain average results, in the Z-axis direction and for all cases mentioned, are plotted in the graph presented in Figure 12.

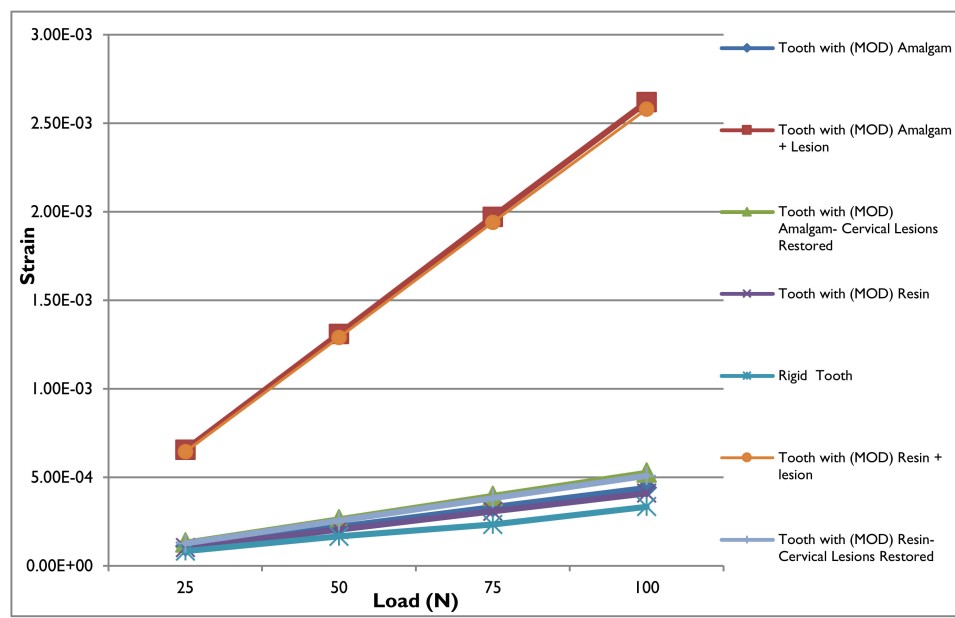

**Figure 12.** Graph of the average results of the tension of the FEM technique in the direction of the Z-axis for all the mentioned cases.

## 2.3. Moiré Projection Analysis

The moiré projection technique is a useful tool in structural engineering for measuring and controlling complex body geometries [17]. The method was applied to strain experimental evaluation of premolar tooth samples under the same conditions mentioned for the FEM analysis. The moiré projection technique is applied according to the system diagram shown in Figure 13a, where the projection system (SP) directs light beams (RP) over the study object (PR-D: tooth), which reflects a network image to the observation system (OS), which captures and superimposes with another network (RO), for the evaluation purpose.

In the implementation of the moiré technique, the same loads used for the simulation of FEM, frontal and oblique loads, were applied in the values of 25, 50, 75, and 100 N.

This technique allows the 3D shape determination by the projection of a regular pattern of straight fringes in an equally spaced pattern after the resulting image processing. The distance between camera and object is 500 mm, the projector lens and camera lens are 110 mm [17]. The best experimental results can be achieved with a small lens and standard projection fringes of 140 lines. This configuration considers the tooth object as the system origin, where the teeth fixing apparatus is called the Originator, which provides good flexibility to define different positions for the object. The equipment used can be considered very common; a Mini Projector Led Benq Gp1 and a CCD camera.

The 3D shape was determined by a technique called "phase shift" which assigns a phase angle to each pixel of the image. There is high uncertainty in the calculation of the phase angle, mainly because the light intensity profile is nearly sinusoidal. This can be mitigated by the acquisition and processing of three or four image fringes.

The software used for data acquisition and processing was FRINGEP, developed specifically for this research and based on the academic version of MATLAB. Considering the pixel size of the camera (CCD) and the pixels on the image plane, the FRINGEP software processes the digital reconstruction of the moiré image, including reconstructions of image intensities, interferogram intensities, phase interferogram, demodulation, and finally, integration to get a 3D representation of the shape and displacement of the deformed tooth.

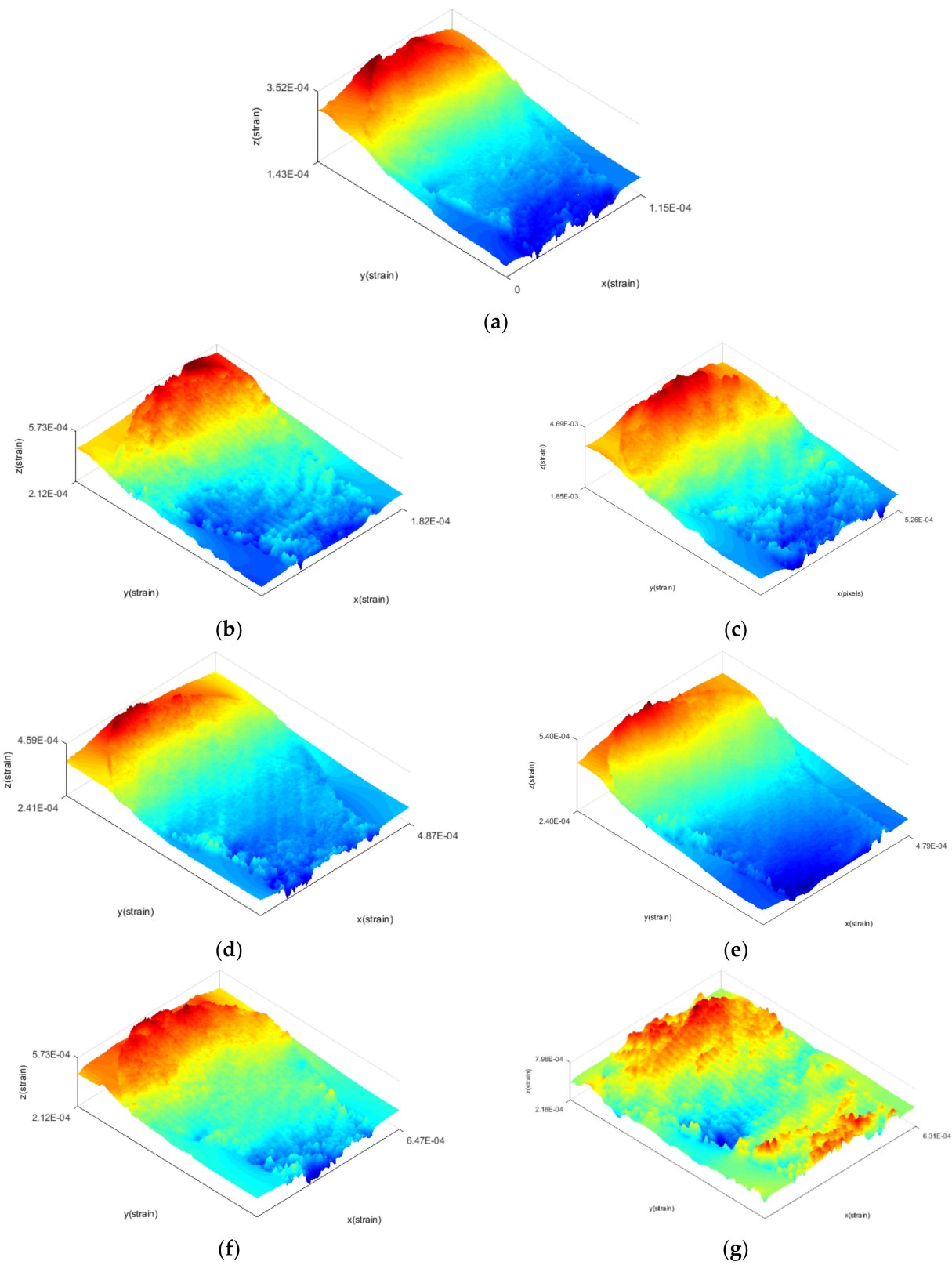

**Figure 13.** *Cont*.

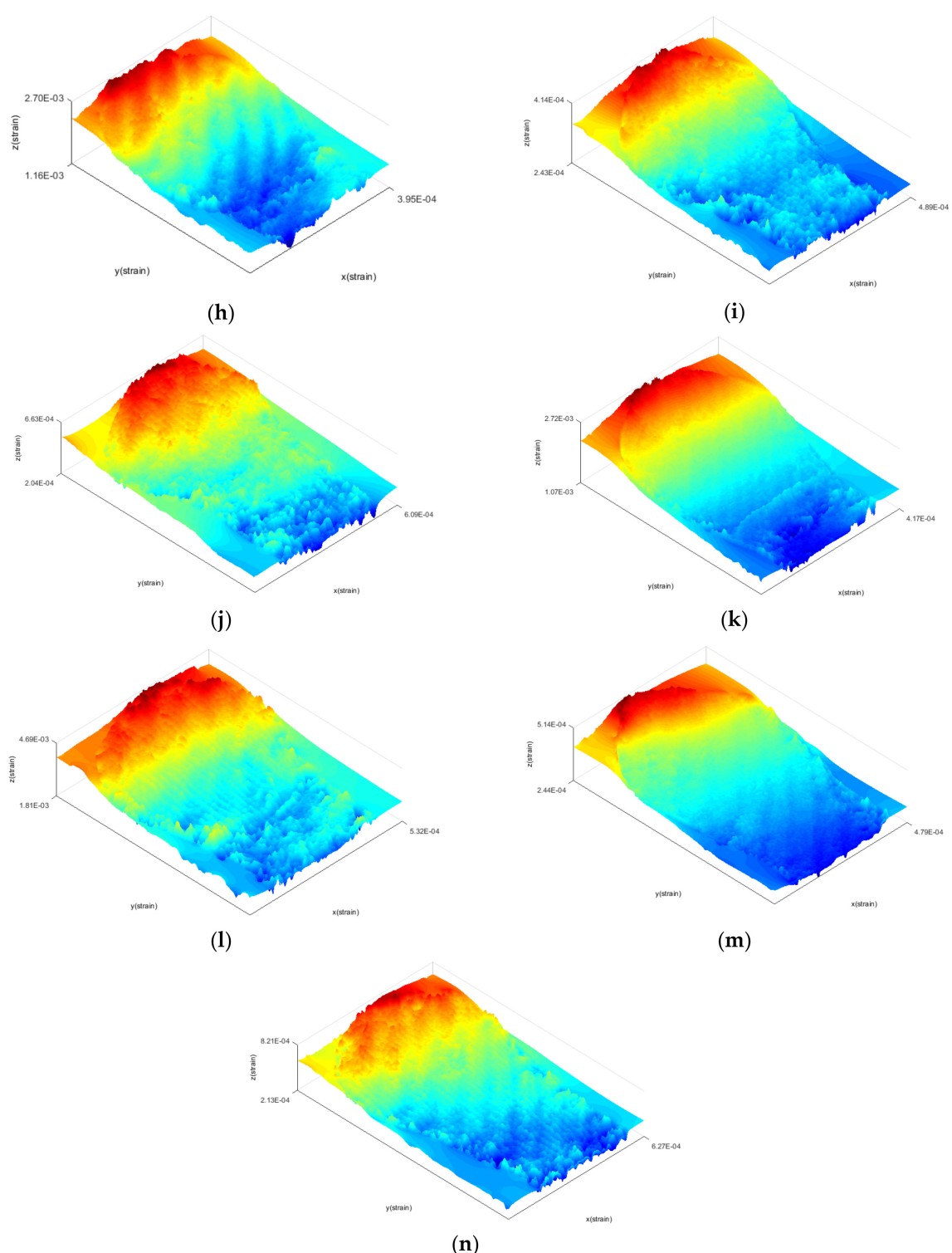

**Figure 13.** 3D color maps representing the strain in the Z-axis direction for 100 N loading: (**a**) tooth H healthy; (**b**) tooth H healthy oblique load 15°; (**c**) tooth AL-MOD amalgam + lesion oblique load 15°; (**d**) tooth A-MOD amalgam; (**e**) tooth ALR-MOD amalgam + restore lesion; (**f**) tooth A-MOD amalgam oblique load 15°; (**g**) tooth ALR-MOD amalgam + restore lesion oblique load. 15°; (**h**) tooth AL-MOD amalgam + lesion; (**i**) tooth R-MOD resin; (**j**) tooth R-MOD resin oblique load 15°; (**k**) tooth RL-MOD resin + lesion; (**l**) tooth RL-MOD resin + lesion oblique load 15°; (**m**) tooth RLR-MOD resin + restore lesion; (**n**) tooth RLR-MOD resin + restore lesion oblique load 15°.

The moiré experimental tests were done for the same specified cases of FEM numerical simulation. The geometric shape and strains were obtained for specified cases. Figure 13 shows, as an example, 3D color maps representing the strain in the Z-axis direction only for 100 N loading.

The experimental moiré projection technique strain average results, in the *Z*-axis direction and for all cases mentioned, are plotted in the graph presented in Figure 14.

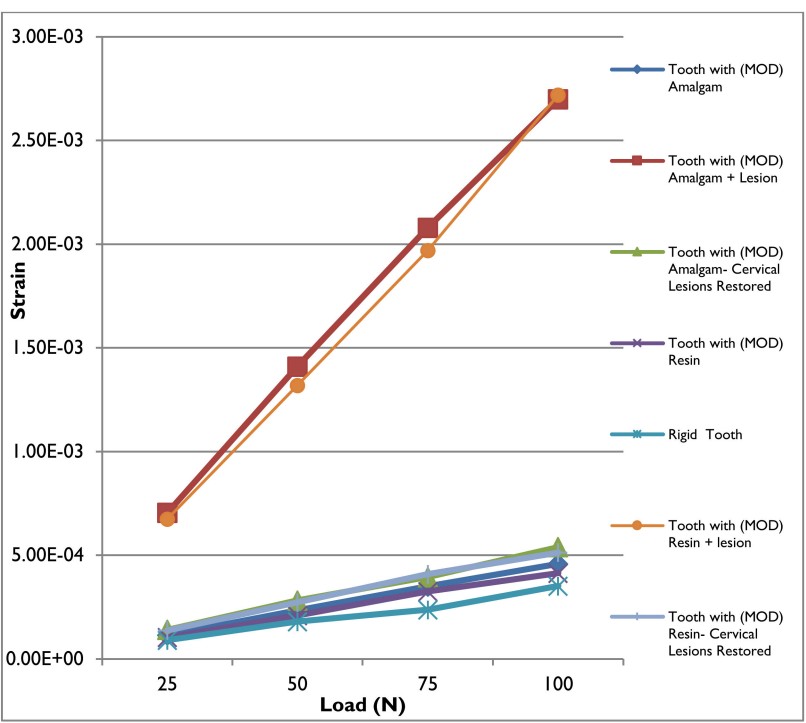

**Figure 14.** Moiré technique strain results in the *Z*-axis direction for all cases analyzed.

## 3. Results and Discussion

An understanding and knowledge of the behavior of healthy and restored teeth submitted to axial loads and tangencies are important in dental preparation planning and is also an indication of restorative material [22].

The analysis of the healthy dental structure was the standard for the analysis of the behavior of the tooth after the preparation in order to verify the areas to be reinforced or altered during the preparation in order to overcome the same possible deficiencies. As the experiments were performed on the same tooth in MEF and moiré, important data were obtained.

From the results obtained through the analysis by finite elements, it was possible to detect that the teeth that present a greater deformation are those where the load was applied at 15°, as can be seen in Figure 11f,l corresponding to tooth AL-MOD amalgam + lesion and tooth RL-MOD resin + lesion, whose values are $415 \times 10^{-6}$ and $410 \times 10^{-6}$, respectively. While, on the other hand, the least deformations occur in the cases of tooth A-MOD-amalgam and tooth R-MOD-resin, with $98 \times 10^{-6}$ and $99 \times 10^{-6}$, as presented in Figure 11c,i, respectively. For the cases of tooth AL-MOD amalgam + lesion and tooth RL-MOD resin + lesion similar behaviors are presented, with values of $211 \times 10^{-6}$ and $207 \times 10^{-6}$, as observed in Figure 11e,k. In the same way, lesions with restorations with different materials are presented, tooth ALR-MOD amalgam + restore lesion and tooth RLR-MOD resin + restore lesion, whose deformation is $100 \times 10^{-6}$ in both cases, according to Figure 11g,m.

From the analysis carried out and from the results obtained using the moiré method, the cases with the greatest deformation are tooth with (MOD) amalgam + lesion and tooth

with (MOD) resin + lesion, both with a value of $2.7 \times 10^{-3}$. In the same way, behaviors are presented in, tooth with (MOD) resin-cervical lesions restored and tooth with (MOD) amalgam-cervical lesions restored, whose maximum values are $5.0 \times 10^{-4}$ and $5.5 \times 10^{-4}$, respectively; as well as the case with tooth (MOD) amalgam and tooth with (MOD) resin, with a maximum deformation value of $4.0 \times 10^{-4}$ for both. All of these results are presented in Figure 14.

The qualitative and quantitative analysis of each test specimen in each of the experimental situations can help in the understanding and evolution of dental preparations and the improvement of restorative materials. These results may indicate the need to change tooth wear during preparation by sparing healthy structures when using adhesive restorations. It is believed that these data may also inform us about some possibility of changing some form of preparation for amalgam.

The study of the reduction of dental structure is the main factor modulating the deformation and resistance to fracture of the dental element. This reduction can occur in different regions of the dental structure, such as inside the dental crown, reducing cusp height, inside the root canal, and in the cervical region [16].

The FEM and moiré techniques are totally non-intrusive, non-destructive, and allow global, non-contact measurement of displacement fields with sub-micrometric resolution. The deformation of the dental structure of each sample was analyzed in different phases and sequentially, that is, each sample was analyzed in the rigid phase without the occlusal box preparation. Afterwards it received sequential wear reducing the height of the axial wall. Each phase is measured separately. The preparations were made with a diamond tip mounted on a standard machine for cavity preparation [16].

The presence of the lesion increased shear values for both restorative materials subjected to axial loading. In oblique loading, there were no significant differences in both resin and amalgam. However, oblique loading increased compliance values with respect to axial loading, mainly for the AL group, since it presented higher strain values when compared to the other groups. The cervical lesions restored with composite and amalgam recovered biomechanical behavior similar to that of uninjured samples.

Some authors describe how under axial loading, the biomechanical behavior of the LA group has higher strain levels as compared with the RL. This can be explained by the amalgam elasticity modulus value, which is lower than the modulus of the composite.

The biomechanical analysis of these endodontically treated premolar teeth restored with amalgam, subjected to oblique loading, and with cervical lesion revealed behavior that could jeopardize the tooth structures and restorative materials. The tooth AL-MOD amalgam + lesion and tooth RL-MOD resin + lesion showed about five times the level of strain, when compared with all the other cases. On the other hand, when the cervical lesions are restored with amalgam or resin, the biomechanical behaviors are similar to those of healthy teeth.

The biomechanical results obtained by FEM and moiré show that the healthy teeth presented a lower level of normal maximum strain in any of the three directions in the *X*-, *Y*-, and *Z*-axis, for all different applied loads. For all analyzed cases, the application of oblique loading induces higher strain levels.

## 4. Conclusions

Even in the geometrically complex structures, such as human upper premolar teeth, the very low displacement and strain levels can be measured by using the moiré projecting technique, as detailed in this study.

The results obtained by both techniques show that healthy teeth presented the lower level of normal maximum strain for all different applied loads. In the meantime, oblique loading induced higher strain levels for all analyzed cases. In addition, it was noticed that the cervical lesions restored with composite and amalgam recovered biomechanical behavior similar to that of uninjured samples.

The results obtained by FEM simulation can be considered perfectly validated by the results presented by the experimental moiré projection analysis, demonstrating that the FEM numerical analysis can be used to evaluate the biomechanical behavior of healthy and endodontic treated teeth. Developing an alternative protocol to generate FEM three-dimensional models will lead to a ready and inexpensive tool since there is no need for costly equipment, for the prognosis of teeth extraction.

**Author Contributions:** Data curation, J.L.V.R., P.V.S. and H.F.L.; Formal analysis, J.O.R.P.; Funding acquisition, H.F.L.; Investigation, F.A.P.; Methodology, G.F.P.R. and H.F.L.; Project administration, G.G.d.P.; Resources, E.G.; Software, G.F.P.R. and H.F.L.; Supervision, J.O.R.P. and M.V.F.; Visualization, G.M.; Writing—original draft, J.L.V.R.; Writing—review & editing, J.O.R.P., M.V.F. and A.R.S. All authors have read and agreed to the published version of the manuscript.

**Funding:** The funds for the execution of this research project come from the Research Support Foundation of the State of Sao Paulo, Brazil.

**Institutional Review Board Statement:** It was selected and collected from the Faculty of Dentistry Bank of Teeth, University of Sao Paulo, twenty one (21) human healthy pre upper molars (1st and 2nd premolars), with indication for extraction. This selection was approved by the Research Ethics Committee: Final Analysis no. 520/11 of the Research Ethics Committee for the Protocol Registration CEP/UFU No. 171/11 and Final Analysis 372/11 of the Research Ethics Committee for the Protocol Registration CEP/UFU 065/11 the Research Ethics Committee of the Federal University of Uberlandia.

**Informed Consent Statement:** Not applicable.

**Data Availability Statement:** Not applicable.

**Conflicts of Interest:** The authors declare no conflict of interest.

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
