# Peer review of "The Restored Premolars Biomechanical Behavior: FEM and Experimental Moiré Analyses"

_applsci, doi:10.3390/app12136768_

Round 1
Reviewer 1 Report
The paper has presented a good study. The results are of a high interest for publishing in Applied Sciences.
Only a few comments required:
The captions of most figures are strangely written. Most figures says "source: Authors". I think there is no need to mention the source unless it is from different author(s).
Figures with sub-plots needs to be labeled in (a),(b),(c),...etc. Only Fig. 13 is correct, others are not.
There is no numbering for the referances.
Author Response
Dear reviewer,
Thanking you for your contribution to the publication of the article, in the attached document you can find the manuscript with the corrections according to your recommendations.
Sincerely,

Reviewer 2 Report
Dear authors,
the article covers a very interesting topic and I support its publication.
However, I suggest some changes in order to improve the overall quality of the manuscript for the readers.
References have no numbers, please add reference number in reference’s list.
Line 53-54.
A shrinkage splitting restorative technique could also be cited, expecially in huge restorations. The authors could add the following reference: “The Simultaneous Modeling Technique: closing gaps in posteriors” PubMed ID 26835524
Line 46-47.
The authors could add a reference to other possible restorative treatments. A sentence that the authors could add could be:
“Other typoes of treatments, generally less conservative, are available to treat posterior teeth, using indirect treatments, nowadays generally performed with CAD/CAM technology.” You could support this sentence with the following reference: External marginal gap variation and residual fracture resistance of composite and lithium-silicate CAD/CAM overlays after cyclic fatigue over endodontically-treated molars DOI 10.3390/polym13173002
Line 122. Please add a sentence to introduce FEM, describing the technique, and cyting some study with its application iin the dental field. For example: Stress distribution in carbon-post applied with different composite core materials: a three-dimensional finite element analysis DOI 10.1080/01694243.2017.1304172
Line 199-204: Please invert the following two sentences, so that the Moiré technique is described first in the paragraph.
“The Moiré Projection method was applied to strain experimental evaluation of 200 premolar teeth samples under the same conditions mentioned for the FEM analysis. The 201 Moiré projection technique is a useful tool in structural engineering for measuring and 202 control complex body geometries, Chengmeng et al. (2017).”
Please add in the conclusions data found on the premolars. Not only advantages or disadvantages or FEM or Moiré analyses.
Please describe the limits of this study design and also possibile future studies to be performed to better understand this topic.
Author Response

(The authors gave the same response as above.)

Reviewer 3 Report
" The restored premolars biomechanical behavior: FEM and experimental Moiré analyses "
It is very interesting to evaluates the influence of different types of restorative material in premolars cavities, associated with the presence of cervical lesions under the action of axial and oblique loads. However, there are a few corrections that are essential to meet the standard for publication. Please refer to the following comments.
1) Figure 13(b) is very blurry. Please replace it with a clear image and add a detailed explanation.
2) Please provide specific numerical values for the qualitative and quantitative analysis of each test specimen in each experimental situation and compare it with the past literature.
3) There are a lot of old citations in your manuscript. Please update to the newest literature as much as possible.
4) Please add your research limitation. Finite element analysis is performed in many fields, please add it in comparison with past literature.
Author Response

(The authors gave the same response as above.)

Round 2
Reviewer 3 Report
Thank you for giving me this opportunity to re-review your revised manuscript.
I am happy that all of the suggested corrections have been made.
Thank you for spending so much time for revised manuscript.